# Dissecting myogenin-mediated retinoid X receptor signaling in myogenic differentiation

Saadia Khilji[1], Munerah Hamed[1], Jihong Chen[2] & Qiao Li [1,2 ✉]

Deciphering the molecular mechanisms underpinning myoblast differentiation is a critical step in developing the best strategy to promote muscle regeneration in patients suffering from muscle-related diseases. We have previously established that a rexinoid x receptor (RXR)-selective agonist, bexarotene, enhances the differentiation and fusion of myoblasts through a direct regulation of MyoD expression, coupled with an augmentation of myogenin protein. Here, we found that RXR signaling associates with the distribution of myogenin at poised enhancers and a distinct E-box motif. We also found an association of myogenin with rexinoid-responsive gene expression and identified an epigenetic signature related to histone acetyltransferase p300. Moreover, RXR signaling augments residue-specific histone acetylation at enhancers co-occupied by p300 and myogenin. Thus, genomic distribution of transcriptional regulators is an important designate for identifying novel targets as well as developing therapeutics that modulate epigenetic landscape in a selective manner to promote muscle regeneration.

[1] Department of Cellular and Molecular Medicine and Faculty of Medicine, University of Ottawa, Ottawa, ON, Canada. [2] Department of Pathology and Laboratory Medicine, Faculty of Medicine, University of Ottawa, Ottawa, ON, Canada. ✉email: Qiao.Li@uOttawa.ca

Recent large-scale functional genomic studies have elucidated the impact of epigenetic control on stem cell fate decision. Histone acetylation is one form of epigenetic regulation that affects DNA accessibility for lineage-specific transcription factors to direct the differentiation of stem cells into distinct lineages, including skeletal myoblasts[1].

The formation of skeletal muscle requires temporal-spatial expression of myogenic regulatory factors (MRFs) that overlap in their ability to convert stem cells into the myoblast lineage[2]. Amongst the MRFs, Myf5 and MyoD are considered as determination factors since they regulate the proliferation and early differentiation of myoblasts, while myogenin functions downstream of MyoD to activate muscle gene expression and is critical for the differentiation and fusion of myoblasts into multinucleated myotubes[2,3]. While Myf5 and MyoD collaborate with partial redundancy in the development of open chromatin formation at muscle-specific loci, myogenin has a unique role in transcription from genes that have been primed[4]. At the structural level, MRFs are a family of basic helix-loop-helix (bHLH) transcription factors that bind to a conserved core hexanucleotide motif (CANNTG) known as E-box[5]. MRF binding is modulated by E-box accessibility, central dinucleotide pair and flanking sequences, which may vary between biological systems and conditions[6–8]. As such, the myogenic program is genetically governed by E-box components, whereas E-box accessibility is controlled epigenetically[6,9].

Genome-wide profiling of histone marks has allowed segmentation of the myoblast genome into distinct chromatin states characterized by a combination of histone modification patterns present in those regions[10]. During myoblast differentiation, an overall decrease in acetylation of H3K9, H3K18, and H3K27[11,12] is accompanied by an increase in H4 acetylation, specifically at MyoD targets[8,12]. Overexpression of myogenin in differentiating C2C12 myoblasts correlates to hyperacetylation of H4, which is associated specifically to late muscle genes[13]. The global decrease in histone acetylation during myoblast differentiation likely reflects a downregulation of proliferation genes, whereas histone acetylation increases at loci important for differentiation. For example, the regulatory regions of Myod and Myf5 are associated with an increase in H3K18 and H3K27 acetylation, connected specifically to the histone acetyl-transferase (HAT) activity of p300[14,15].

Initially identified as an E1A-associated protein[16], p300 is a critical transcriptional co-activator of a myriad of transcription factors involved in many cellular processes including proliferation and differentiation[17]. It instigates chromatin remodeling as a HAT, but acts also as a molecular scaffold, bridging different DNA-binding proteins and activators with the basal transcriptional machinery[18–20]. Particularly, H3K18 and H3K27 are the acetylation targets of p300 prior to RNA polymerase II recruitment[21]. While p300 can be found at the promoter regions[22], its occupancy is regarded as the best chromatin signature of enhancers[23,24] which can be further categorized as poised, marked by H3K4me1/2, or active, signified by H3K27ac in addition to H3K4me1/2[25,26]. In contrast, H3K27me3 is tightly associated with inactive genes and has a predictive nature of gene silencing[27]. We have recently utilized different histone marks within the proliferating myoblasts to generate a 14-state chromatin state model to characterize loci-specific histone acetylation at p3000-associated enhancers in early myoblast differentiation, particularly when it is recruited by MyoD[12]. Our data presents a model of histone acetylation increases at distinct genomic loci despite a global decrease in histone acetylation as myoblasts differentiate[12].

Nuclear receptor superfamily of transcription factors regulate gene expression in response to steroids, lipids, and other small molecule ligands[28]. As a member of this family, retinoid X receptor (RXR) binds to DNA either as a homodimer or heterodimer with other family members, making it a partaker of a large array of signaling pathways and a significant modulator of drug targets. We have previously reported that a RXR-selective agonist, bexarotene, enhances the differentiation and fusion of myoblasts through the function of RXR as a transcription factor[29]. In addition, this enhancement is mediated largely through a direct regulation of MyoD gene expression and coupled with an augmentation of myogenin protein[10]. While MyoD is required for early myogenic differentiation, the activation of a subset of late muscle-specific genes occurs most efficiently in the presence of myogenin that amplifies the expression of genes previously primed by MyoD[30].

Nonetheless, the molecular pathways of transcription partners, co-activators and chromatin state dynamics underlying myogenin-mediated myoblast differentiation in response to RXR signaling remain an important but poorly understood issue. Here, we delineate how RXR-selective signaling affects genome-wide myogenin binding characteristics and dissect the molecular pathways through which RXR-selective signaling promotes myoblast differentiation.

## Results

**Genome-wide localization of myogenin in RXR signaling**. We have previously reported that a RXR-selective agonist, bexarotene, enhances the differentiation and fusion of skeletal myoblasts into mature myotubes, while augmenting the protein level of myogenin, a terminal differentiation factor[10,29]. To study the molecular pathways underlying myogenin-mediated myoblast differentiation in the context of RXR signaling, we examined genome-wide myogenin localization using chromatin immunoprecipitation coupled with deep-sequencing (ChIP-seq), in the well-established myoblast model[11,31,32]. As shown in Supplementary Fig. 1a, b, the expression of myogenin was induced upon 24 h of differentiation, which was significantly augmented furthermore by over threefold following the addition of bexarotene. The myoblasts were subsequently subjected to myogenin ChIP-seq experimentation.

The quality metrics of myogenin ChIP-seq data fell within the high confidence range for base quality, as represented by the average base quality score for myoblasts differentiated with or without bexarotene (score of 30–40, Supplementary Fig. 1c). Fingerprint analysis with the deepTools suite showed an enriched but localized ChIP-seq read signal, 10% of the myoblast genome was enriched with 70% of uniquely aligned reads (Supplementary Fig. 1d). Additionally, 98% of reads were successfully aligned to the mm9 genome with an IP efficiency of 2.9% and 3.1% for myoblasts differentiated with or without bexarotene, respectively, as projected by Hypergeometric Optimization of Motif EnRichment (HOMER) program analysis. Following peak calling of the sequencing data, about 9500 confident myogenin peaks were obtained across both conditions (Fig. 1a), where the center of peaks displayed higher phylogenetic conservation (PhastCons) scores than the surrounding regions, signifying well-aligned and ChIP-ed binding sites (Supplementary Fig. 1e). In addition, Integrated Genome Viewer (IGV)[33] of myogenin read signals displayed a distinct enrichment at myogenic loci (Fig. 1b), including regions flanking ADORA1 loci implicated previously in skeletal muscle injury and repair[34–36].

**RXR signaling directs myogenin to poised enhancers**. When categorizing the events of myogenin binding, we found approximately 55% of the peaks from myoblasts differentiated with bexarotene to be unique to bexarotene (Fig. 1a), although the

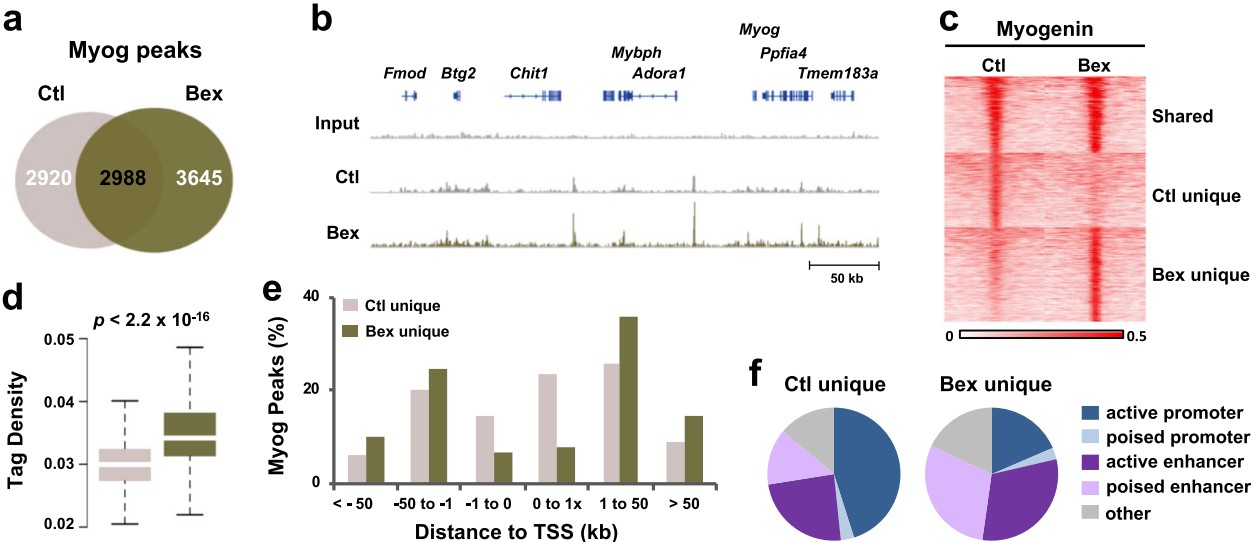

**Fig. 1 Characterization of myogenin localization in response to RXR signaling. a** Venn diagram depicts the overlap of myogenin ChIP-seq peaks between C2C12 myoblasts differentiated in the absence or presence of bexarotene (Ctl or Bex) for 24 h. **b** Myogenin ChIP-seq read signals surrounding the *Adora1* locus. **c** Myogenin signal intensity was centered around myogenin peaks (±1 kb), categorized into sites that were unique to control or bexarotene, and those shared between the two conditions. **d** Quantification of peak tag density (total number of tags in the peak region/ length of the peak) for the myogenin peaks (Wilcoxon rank sum test). **e** Distribution of peak distance to the nearest TSS. **f** Myogenin peaks were associated with distinct chromatin states.

length of peaks was similar between bexarotene and control conditions (Supplementary Fig. 1f). There was a distinct increase in the number of myogenin peaks unique to bexarotene (3645) in comparison with unique to control (2920). In addition to the increase in the number of peaks, there was an increase in peak signal intensity in response to RXR signaling, as demonstrated by the heatmaps (Fig. 1c). Quantification of tag density from each category showed that myogenin peaks unique to bexarotene consist of a significantly greater number of tags per peak compared with peaks unique to control (Fig. 1d).

Given that myogenin is mainly associated with promoter regions[10,32], we analyzed if RXR signaling affects the distance of myogenin peaks to the nearest transcription start site (TSS). Comparing to 40% of myogenin loci unique to control, only 15% of loci unique to bexarotene were found to be within 1 kb of a promoter (Fig. 1e). As result, the majority of loci unique to bexarotene were located distal to the promoters (>1 kb), suggesting that RXR signaling yields a preference for myogenin to bind regions likely to be enhancers (Fig. 1e). We also utilized an established chromatin state model based on genome-wide co-occurrence of different epigenetic marks in proliferating myoblasts[10], to classify the chromatin state distribution of myogenin loci in differentiation. As previously reported[10,32], myogenin loci unique to control were largely associated to active promoters (45%, Fig. 1f), in line with the close TSS proximity of a large proportion of myogenin peaks (Fig. 1e). However, in response to RXR signaling, the proportion of myogenin loci falling within enhancer regions increased from 38 to 60%, and especially to poised enhancers where myogenin association doubled from 13 to 30% (Fig. 1f). In contrast, the association of myogenin to active promoters diminished from 45 to 19% (Fig. 1f). Thus, a genome-wide shift in myogenin association from promoter to enhancer regions may be a functional signature of RXR signaling in myogenic differentiation.

**RXR signaling promotes myogenin binding preference**. To determine if RXR signaling affects myogenin binding site preferences, we performed de novo motif analysis on myogenin loci unique to bexarotene or control, covering sequences ±50 bps to

the center of peaks. The most enriched and significant primary motif across both categories was a canonical CANNTG E-box (Fig. 2a). Although E-box was the most significant motif identified across both categories, ~63% of myogenin loci unique to bexarotene harbored an E-box compared with only 21% of loci unique to control (Fig. 2a). Additionally, quantification of motif enrichment revealed that the E-box was enriched more than fourfold at myogenin loci unique to bexarotene, compared with loci unique to control, while little changing in the enrichment of other identified motifs (Supplementary Fig. 2).

Interestingly, while myogenin loci from both categories harbored an E-box with a GC at its core, E-box unique to bexarotene displayed a lower degree of consensus at the central G nucleotide (Fig. 2a), reflecting possibly an increased adaptability of myogenin recognition in response to RXR signaling. Furthermore, distinct from unique to control, E-box unique to bexarotene contained two additional consensus flanking nucleotides (Fig. 2a), potentially increasing the stringency of myogenin recognition upon RXR signaling, since nucleotides outside the E-box determine binding specificity by influencing the three-dimensional structure of DNA binding sites[37].

To further understand the impact of RXR-selective signaling on myogenin binding site preferences in early differentiation, we analyzed publicly available myogenin ChIP-seq data from C2C12 myoblasts differentiated for 24, 60 h, and 7 days. De novo motif analysis revealed that an increasing proportion of myogenin loci (from 45 to 77%) harbor the E-box motif as differentiation proceeds from 24 h to 7 day, similar to myoblasts differentiated with bexarotene for 24 h (Fig. 2a, Supplementary Fig. 3a). We also analyzed myogenin ChIP-seq data from primary myoblasts isolated from gastrocnemius of C57BL/6 mice and differentiated for 24 h[38]. Interestingly, the central and flanking nucleotides of the E-box bound by myogenin in differentiating primary myoblasts were most similar to C2C12 myoblasts differentiated for 7 day or with bexarotene for 24 h (Fig. 2a, Supplementary Fig. 3a). In addition, it is only in primary myoblasts differentiated for 24 h or C2C12 myoblasts for 7 day, we discerned two additional consensus flanking nucleotides, mirrored by myoblasts differentiated with bexarotene for 24 h (Fig. 2a, Supplementary Fig. 3a).

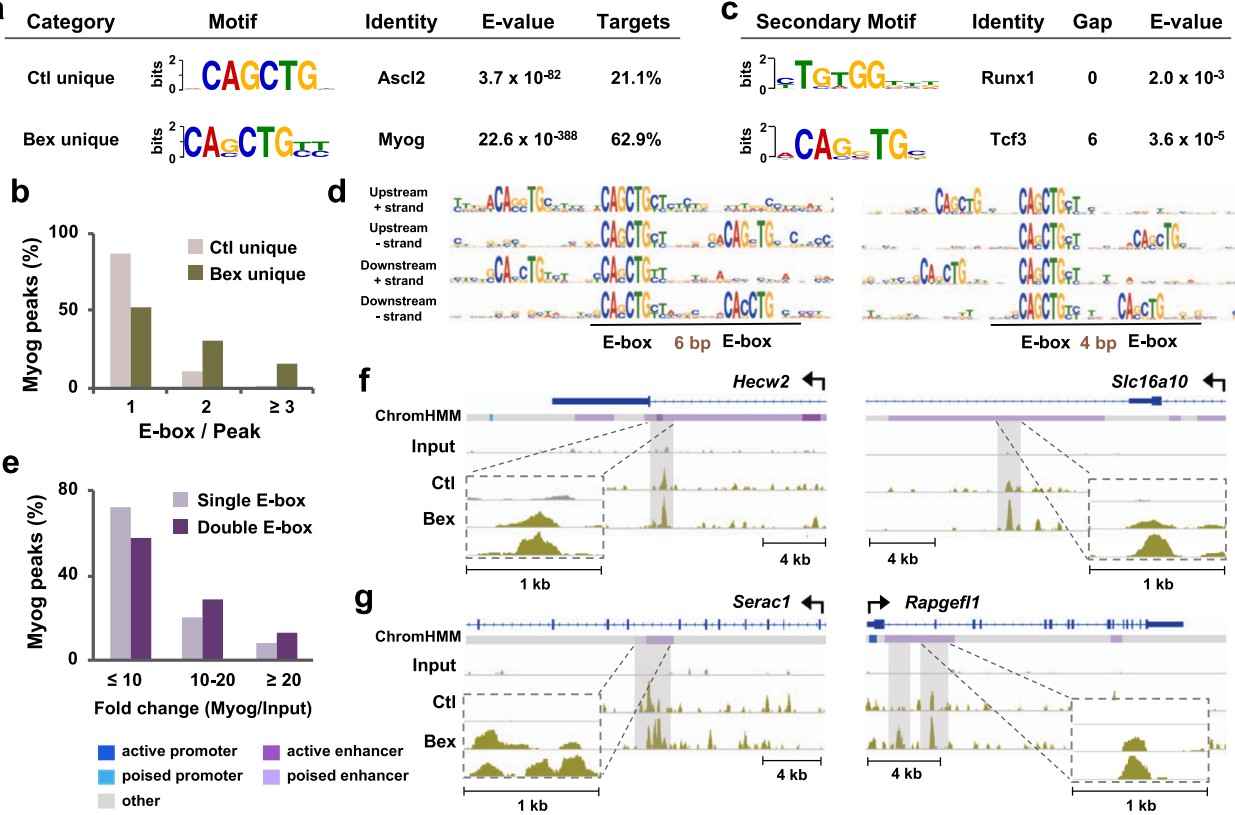

**Fig. 2 RXR signaling influences myogenin binding site specificity. a** The most significant and centrally distributed consensus motif, its associated identity, E-value, and the percentage of myogenin loci associated to each motif as revealed by *de novo* motif analysis of myogenin peaks unique to control or bexarotene (Ctl or Bex) following 24 h of differentiation. **b** Bar graph displays the distribution of myogenin peaks associated to the indicated number of E-box motifs. **c** The predicted secondary motif, its associated identity, E-value, and base pair spacing between the primary and secondary motifs projected by MEME secondary motif analysis. **d** Presentation of the primary and secondary motifs for myogenin loci unique to bexarotene. **e** Myogenin enrichment over input for myogenin peaks unique to bexarotene, categorized as those associated to single E-box motifs or double E-box motifs. **f**, **g** Genome browser view of *Hecw2*, and *Slc16a10* with myogenin peaks associated to single E-box, and *Serac1* and *Rapgefl1* with myogenin peaks associated to double E-box. Dashed boxes show a zoomed in view of the regions boxed in gray. Blue bars show Refseq gene positions.

Next, we quantified the number of E-box motifs per myogenin peak, and found that 47% of myogenin loci in myoblasts differentiated with bexarotene for 24 h contained 2 or more E-boxes similar to 58% and 43% of myogenin loci in 7-day C2C12 and 24-h primary myoblast differentiation respectively (Fig. 2b, Supplementary Fig. 3b). In contrast, only 13% of myogenin loci unique to control contained 2 or more E-boxes per peak, comparable to the public data from myoblasts differentiated for 24 and 60 h (Fig. 2b, Supplementary Fig. 3b). Quantification of motif density also revealed greater E-box density at myogenin peaks from 7-day C2C12 and 24-h primary myoblast differentiation than myoblasts differentiated for 24 and 60 h (Supplementary Fig. 3c). Thus, RXR signaling modulates myogenin binding preferences in terms of E-box identity and density, reflecting a dynamic environment representative of an ex vivo context and mature myotubes.

As gene expression is often controlled by regulatory circuits consisting of multiple transcription factors, we explored regions proximal to the primary E-box for neighboring transcription factor motifs using MEME secondary motif analysis. Runx1 motif was the most significant motif associated with E-box unique to control (Fig. 2c), consistent with previous observation that Runx1 cooperates with MRFs and the AP-1 family of transcription factors to regulate myoblast proliferation and differentiation[38]. However, among myogenin loci unique to bexarotene, we identified a double-E-box, two canonical E-boxes, separated by 4 or 6 bp (Fig. 2c, d). Previous characterization of double E-box

has modeled a spacing of 5 bp as optimal for a full turn of the DNA double helix, such that the nucleotides in each E-box face the same spatial direction of DNA[39]. Taken together, our results suggest that RXR signaling not only modulates the preference of myogenin for E-box component, but may also promote its cooperation with flanking E-box binding protein for complex formation.

Next, we discerned in detail the population of myogenin loci unique to bexarotene containing a single E-box or double E-box. While about 28% of single E-box associated loci displayed a fold change greater than 10 in the enrichment of myogenin over input, ~42% of double E-box peaks fell into this category (Fig. 2e). Moreover, myogenin loci associated with single E-box generally appeared as sharper and narrower peaks compared with a broader compilation of read signals at loci containing the double E-box, shown by IGV snapshots of representative myogenin ChIP-seq tracks (Fig. 2f, g). Taken together, E-box components in central dinucleotide, flanking sequences and density, as well as flanking motif composition may all contribute to myogenin binding as a functional avenue of RXR signaling in myogenic differentiation.

**Epigenetic features associated to myogenin loci.** As transcription factor occupancy is an important indicator of lineage-specific gene expression, we integrated genome-wide myogenin loci with differential RNA-seq of myoblasts in matching conditions[10]. To this end, genes upregulated between proliferation and differentiation were subdivided into groups affected by differentiation

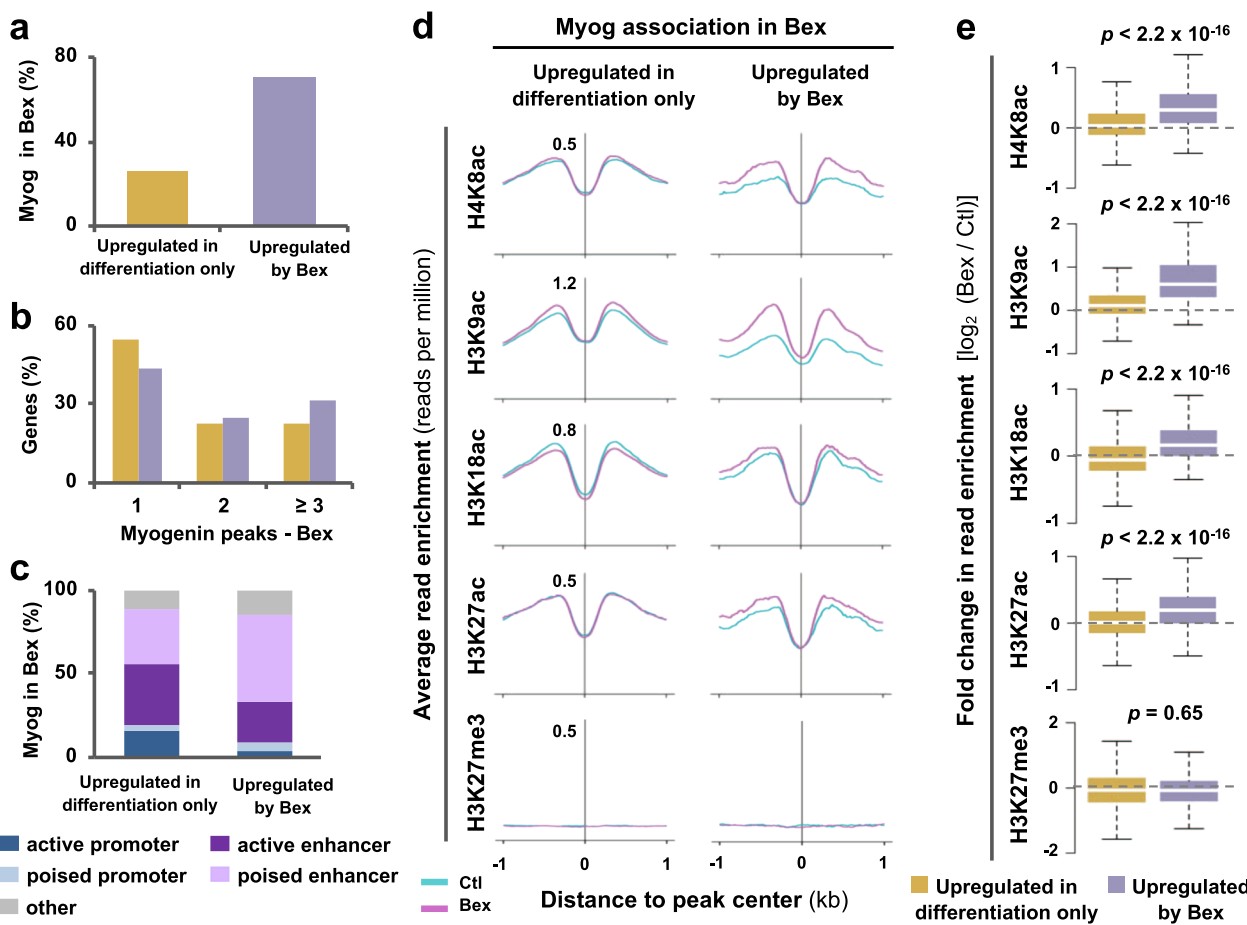

**Fig. 3 Myogenin is associated with bexarotene-responsive gene expression. a** Myogenin loci from myoblasts differentiated with bexarotene (Bex) for 24 h were associated with genes upregulated in differentiation only or additionally by bexarotene. **b** Quantification of the number of peaks per gene. **c** Chromatin state distribution of myogenin loci. **d** Average read density for acetylation of H4K8, H3K9, H3K18, and H3K27 in addition to H3K27 trimethylation from myoblasts differentiated in the absence (Ctl) or presence of bexarotene. **e** Quantification of log2-fold difference in the histone acetylation and methylation read signal at myogenin loci categorized in (**d**) (Wilcoxon rank sum test).

only or bexarotene additionally, and each group of genes was then examined for myogenin association in response to RXR signaling. Approximately 25% of genes upregulated in differentiation only were associated to myogenin, whereas about 70% of bexarotene-responsive genes displayed an association to myogenin (Fig. 3a). Furthermore, 57% of bexarotene-responsive genes were associated with two or more myogenin peaks compared with 45% of genes in differentiation only (Fig. 3b). Taken together, our results suggest that myogenin is intimately connected to rexinoid-responsive gene expression in early myoblast differentiation.

Since RXR signaling modifies genomic localization of myogenin (Fig. 1g), we explored the chromatin state distribution of myogenin associated with genes upregulated in differentiation only or by bexarotene additionally. Approximately 53% of myogenin loci associated with bexarotene-responsive genes were found in poised enhancers, compared with about 33% of myogenin loci associated with genes upregulated in differentiation only (Fig. 3c). Since histone acetylation and methylation are important for gene expression, we also analyzed H4K8ac, H3K9ac, H3K18ac, and H3K27ac centered at myogenin loci associated with each group of genes. H3K27me3, associated with transcriptional repression[27], was used as control (Supplementary Fig. 4). Interestingly, myogenin loci associated to rexinoid-responsive genes displayed an overall increase in histone acetylation following addition of bexarotene, in contrast to loci associated to genes upregulated in differentiation only where no

change in histone acetylation was observed (Fig. 3d, e). Moreover, the largest increase in histone acetylation in response to bexarotene was H4K8ac and H3K9ac, whereas H3K27me3 signal was minimum (Fig. 3d, e). Therefore, myogenin-mediated RXR signaling may be reflected by residue-specific acetylation in bexarotene-responsive gene expression.

**Genomic distribution of p300 in response to RXR signaling.** Since myogenin-mediated rexinoid-responsive gene expression is associated with a histone acetylation signature and p300 is a critical HAT required for skeletal muscle development[20,40,41], we next examined in detail p300 and histone modification in early myoblast differentiation. Interestingly, the global levels of cellular p300 protein and histone modifications were relatively stable in differentiation and in the presence of bexarotene (Fig. 4a, b). We next conducted p300 ChIP-seq in matching conditions of myogenin ChIP-seq to dissect the role of p300 and residue-specific histone acetylation in response to RXR signaling. As shown in Fig. 4c, the proportion of p300 peaks unique to bexarotene was considerably less than the 14,849 peaks unique to control or the 7198 peaks shared regardless of treatment. When assessing the chromatin state distribution of p300, we found that p300 loci unique to control mainly associated to enhancers with only 20% to the promoter regions (Fig. 4c), similar to previous reports[24,25]. The addition of bexarotene, however, led to a distinct change in

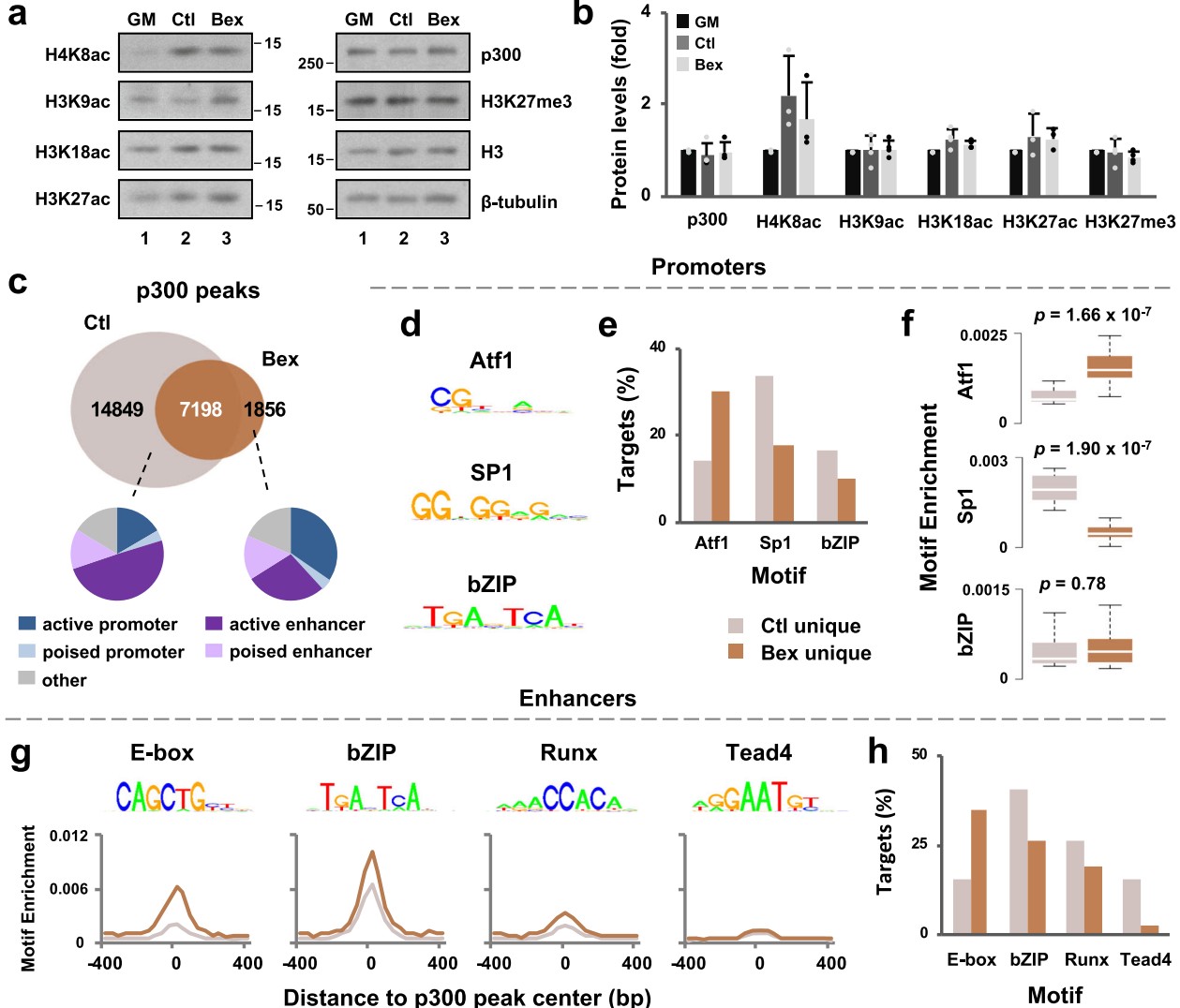

**Fig. 4 Genomic distribution of p300 in RXR signaling. a** Western analysis of p300, and indicated histone marks with β-tubulin as a loading control in proliferating C2C12 myoblasts (GM) and myoblasts differentiated in the absence or presence of bexarotene (Ctl or Bex) for 24 h. **b** Quantification of the Western blots is presented as fold change in relation to proliferating myoblasts (error bars: SD; $n = 3$). **c** Venn diagram depicts unique and shared p300 loci between myoblasts differentiated without or with bexarotene. Chromatin state distribution of p300 loci unique to each condition is also shown. **d** De novo motif analysis of p300 loci in the promoter regions. **e** The percentage of p300 loci harboring motifs displayed in (**d**). **f** Boxplots represent the enrichment of each motif shown in (**d**) (Wilcoxon rank sum test). **g** De novo motif analysis of p300 loci in the enhancer regions with the corresponding motif enrichment. **h** The percentage of p300 loci harboring the motifs from (**g**).

chromatin state distribution with nearly 40% of p300 loci unique to bexarotene found in the promoter regions (Fig. 4c).

As p300 displayed a preference for promoter association in response to RXR signaling (Fig. 4c), we examined motif composition underlying p300 loci for which active and poised promoters as well as active and poised enhancers were simply grouped into promoters and enhancers. De novo motif analysis of p300 associated promoters revealed the presence of binding motifs for Atf1, Sp1, and bZIP transcription factors (Fig. 4d–f). Binding motif for Sp1, a ubiquitously expressed transcription factor involved in the formation of pre-initiation complex (PIC) in communication with coactivators including p300[24,42,43], was particularly enriched in unique to control (Fig. 4e, f). However, bexarotene promoted an increase in the Atf1 motif (Fig. 4e, f), suggesting RXR signaling may promote the distribution of p300 to promoters as well as affect the identity of the recruiting factor.

In contrast to enrollment of p300 at the promoters, p300 occupancy at the enhancers correlated not only with binding

motifs for bZIP transcription factor, but also for E-box binding proteins, Tead4, and Runx (Fig. 4g, h). In addition, E-box enrichment increased over twofold in response to bexarotene, and it was the only motif to which p300 association increased in RXR signaling (Fig. 4g, h). Additionally, nearly 40% of p300 loci unique to bexarotene were associated to an E-box in contrast to only 16% of loci unique to control (Fig. 4h), suggesting that RXR signaling promotes the recruitment of p300 by E-box binding proteins to the enhancers. Taken together, p300 is differentially recruited to promoters and enhancers in that p300 may mediate RXR signaling through distinct functional modes.

**Histone acetylation associated with bexarotene-responsive genes.** Previous studies have found that the level of histone acetylation at promoters decreases during myoblast differentiation regardless if gene expression is up- or downregulated[11,36]. We thus examined the profiles of H4K8ac, H3K9ac, H3K18ac and

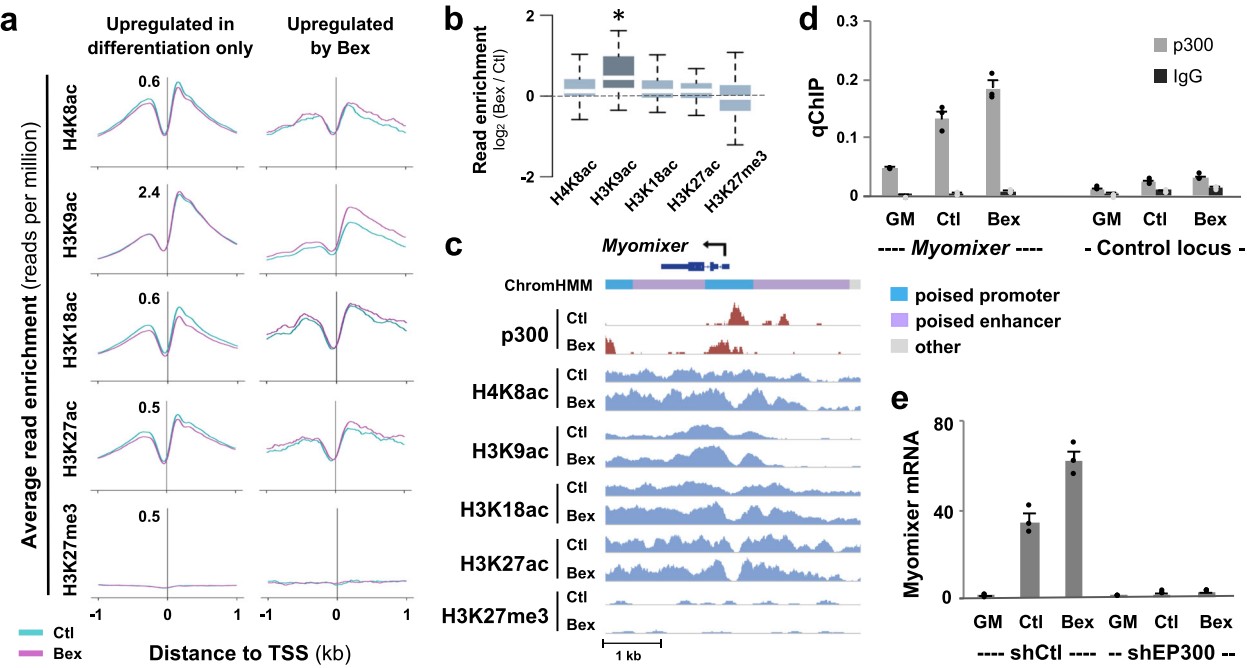

**Fig. 5 Bexarotene-responsive gene promoters are marked by residue-specific histone acetylation. a** Average enrichment profiles of H4K8ac, H3K9ac, H3K18ac, H3K27ac, and H3K27me3 from myoblasts differentiated in the absence or presence of bexarotene (Ctl or Bex) for 24 h. The profiles span 2 kb across the TSS of genes upregulated in differentiation only and by bexarotene additionally. **b** Boxplots present fold change in enrichment of histone modifications between treated and control myoblasts at bexarotene responsive promoters shown in (**a**) (*, $p \leq 0.05$, Wilcoxon rank sum test). **c** IGV view of p300 and indicated histone marks at the *Myomixer* locus. Blue bars show Refseq gene position. **d** Enrichment of p300 at the *Myomixer* promoter was determined by qChIP analysis in proliferating (GM) and differentiating myoblasts. Normal IgG antiserum and a random locus were used as control. Quantification is presented as percentage of enrichment in relation to input chromatin DNA (error bars: SEM; $n = 3$). **e** RT-qPCR analysis of *Myomixer* gene expression following the introduction of p300 shRNA (shEP300). Quantification is presented as fold change relative to proliferating myoblasts, after normalization to an internal control. A nonsilencing shRNA (shCtl) was used as control (error bars: SEM; $n = 3$).

H3K27ac, with H3K27me3 as control, across the promoters of bexarotene-responsive genes. Interestingly, the average signal of H3K9ac at the TSS of bexarotene-responsive genes was significantly enriched following bexarotene treatment (Fig. 5a, b). However, there was no such enrichment observed at the TSS of genes upregulated in differentiation only (Fig. 5a, b). Additionally, there was little detectable H3K27me3 signal at the promoters (Fig. 5a, b). Evidently, the increase in the level of H3K9ac at the TSS of bexarotene-responsive genes was correlated to the transcriptional activity of those genes, suggesting that residue-specific histone acetylation at distinct promoters may play a key role in rexinoid-responsive gene expression.

Since rexinoid-responsive gene expression is linked to residue-specific histone acetylation, we examined the role of p300 in the regulation of myomixer, a microprotein essential for myoblast fusion and muscle formation[44–46], along with the aforementioned histone acetylation. IGV track of ChIP-seq read signal coverage displayed a constitutive presence of p300 at the promoter, accompanied by residue-specific histone acetylation, particularly H3K9ac (Fig. 5c). ChIP-qPCR analysis demonstrated that p300 occupancy at the myomixer locus was enriched by about twofold upon 24 h of differentiation, and further increased following bexarotene treatment (Fig. 5d). We also used a previously established p300 shRNA knockdown myoblast system[20] to determine the requirement of p300 for myomixer expression. As shown in Fig. 5e, in shRNA control myoblasts, myomixer expression was markedly induced in differentiating myoblasts and augmented further by bexarotene. However, shRNA knockdown of p300 attenuated myomixer gene expression in differentiating myoblasts regardless of treatment, suggesting a critical role for p300 in myomixer gene regulation.

**Genomic overlap of p300 with myogenin**. Since p300 can be recruited by E-box binding proteins (Fig. 4), we analyzed the overlap of myogenin loci in differentiating myoblasts with p300 loci in promoters or enhancers unique to control or bexarotene. As shown in Fig. 6a, ~5% of p300 loci unique to control displayed an overlap with myogenin at either promoters or enhancers. Although the addition of bexarotene did not affect the degree of p300 and myogenin overlap at the promoters, their overlap increased from 5 to 23% at the enhancer regions (Fig. 6a). Furthermore, it was only in response to RXR signaling, enhancers associated with both myogenin and p300 displayed a significant increase in H4K8ac and H3K9ac (Fig. 6b, c). Figure 6d shows an IGV snapshot of the sarcomeric gene *Mybph* locus, where the enrichment of p300 and myogenin increased at putative enhancer regions following the addition of bexarotene. As demonstrated by qChIP analysis, myogenin and p300 enrichment was evident at the putative enhancer in differentiating myoblasts and further increased markedly by bexarotene (Fig. 6e). The enrichment of p300 and myogenin at the *Mybph* locus also correlated with an activation of Mybph gene expression as revealed by RT-qPCR analysis in matching conditions (Fig. 6f). Moreover, differentiation-dependent and bexarotene-responsive Mybph gene expression was attenuated following shRNA p300 knockdown (Fig. 6f), designating *Mybph* as a p300 target gene and signifying a role for p300 in mediating rexinoid signaling.

Taken together, our results suggest that RXR signaling modulates the global binding dynamics and motif recognition of myogenin, an important MRF, which cooperates with p300, in the installation of an epigenetic signature associated to rexinoid-responsive gene expression in early myogenic differentiation (Fig. 7).

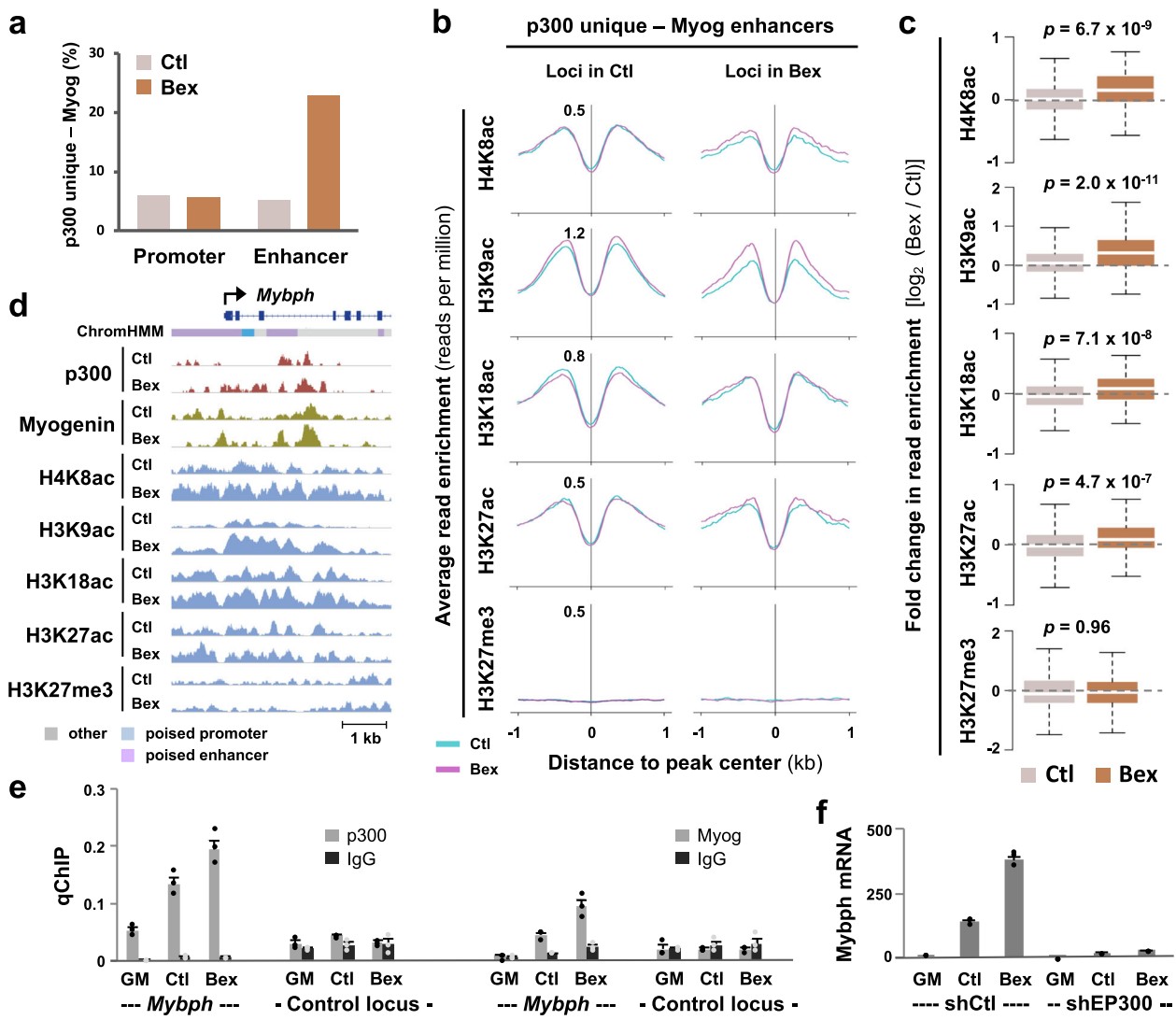

**Fig. 6 Overlap between p300 and myogenin in RXR signaling. a** p300 loci unique to control or bexarotene (Ctl or Bex) were associated to promoters or enhancers and co-localized with myogenin loci in matching conditions. **b** Average read signal profiles for indicated histone marks at enhancers associated with p300 and myogenin. **c** Boxplots present the fold change in enrichment between control and bexarotene for histone modifications shown in (**b**) (Wilcoxon rank sum test). **d** IGV view of p300, myogenin and histone modification read density at the *Mybph* locus. Blue bars show Refseq gene position. **e** Occupancy of p300 at the *Mybph* locus was determined by qChIP analysis in proliferating myoblasts (GM) and myoblasts differentiated for 24 h. Normal IgG antiserum and a random locus were used as controls. Quantification is presented as percentage of enrichment in relation to input chromatin DNA (error bars: SEM; $n = 3$). **f** RT-qPCR analysis of *Mybph* gene expression following the introduction of p300 shRNA (shEP300). A nonsilencing shRNA (shCtl) was used as control. Quantification is presented as fold change relative to proliferating myoblasts normalized to an internal control (error bars: SEM; $n = 3$).

## Discussion

We have previously established that a selective RXR agonist, bexarotene, enhances the differentiation and fusion of myoblasts, while augmenting myogenin expression[29]. Here, we delineate myogenin-mediated RXR signaling in early myoblast differentiation using OMICs approaches. We have found that RXR signaling leads to myogenin occupancy at poised enhancers and a specific "double E-box" with consensus flanking sequences. We have also found a close association of myogenin with rexinoid-responsive gene expression. Furthermore, we have identified an epigenetic signature related to p300 and rexinoid-responsive promoters in connection with residue-specific histone acetylation. Thus, RXR signaling-promoted genomic distribution of transcription regulators presents distinct dynamics which may be utilized to identify novel genetic targets, as well as the potential of pharmaceutical modification of the epigenetic landscape to promote muscle development and regeneration.

Characterization of genome-wide transcription factor distribution and binding motifs is an important avenue for understanding regulatory networks that govern stem cell differentiation. Using a ChIP-seq approach, we have found that RXR signaling shifts the distribution of myogenin from promoter to poised enhancer regions (Fig. 1). As poised enhancers play a crucial role within gene programs that remain in an inactive state until onset of differentiation[26], myogenin may contribute to the initiation of these pathways to elicit myoblast differentiation in response to RXR signaling. Interestingly, myogenin loci unique to rexinoids have a greater association to a distinct E-box with two flanking nucleotides consisting of CASCTGYY, whereas loci unique to control are associated with a shorter motif of CAGCTG (Fig. 2). DNA sequences flanking the core motif exert an impact on transcription factor binding, as they affect DNA topology and as a result binding affinity[47,48]. For example, yeast bHLH binding specificity is determined by nucleotides outside the E-box motifs

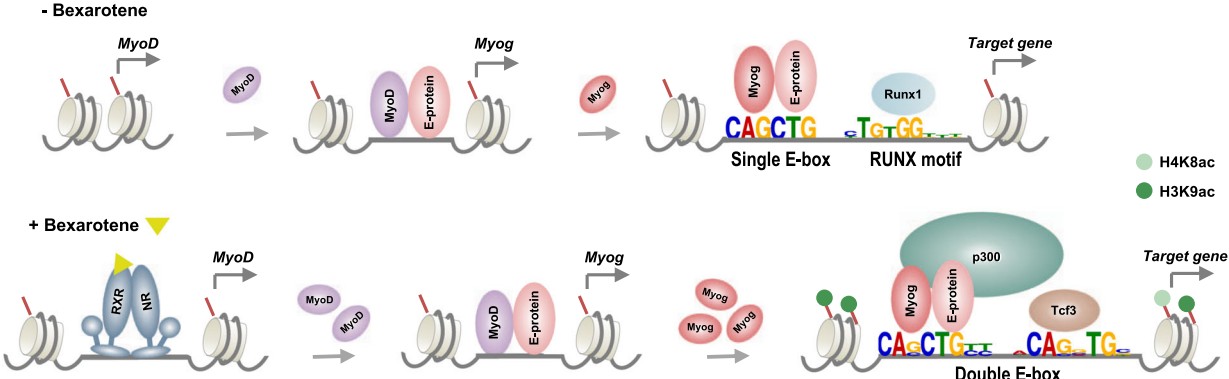

**Fig. 7 A model of the regulatory events underlying myogenin-mediated RXR signaling in myoblast differentiation.** In the absence of rexinoid, a single canonical E-box with RUNX as the most significant neighboring motif is found in myogenin loci. RXR signaling augments directly MyoD gene expression, which in turn enhances the expression of myogenin. Nevertheless, RXR signaling promotes myogenin binding preference to a distinct double E-box motif with a lower degree of consensus at the central G nucleotide but two additional consensus flanking nucleotides. In addition, p300 occupancy as well as an increase in H3K9 and H4K8 acetylation overlap with rexinoid-responsive myogenin loci, important to myogenin-mediated RXR signaling in myoblast differentiation.

due to 3D structure of the DNA binding sites[37]. Thus, the flanking sequences of E-box represent an important mechanism in loci selection of nearly 63% of myogenin occupancy in rexinoid-enhanced myoblast differentiation (Fig. 2). Nonetheless, myogenin binding may also be influenced by factors beyond consensus sequences including chromatin accessibility, DNA topology, protein interactions, and epigenetic regulation such that characterizing consensus motifs is an important first step to further understand gene regulatory networks[49,50].

Apart from the primary motif, RXR signaling also favors a secondary motif proximal to the primary E-box. Runx1, a well-known component of regulatory complexes within the muscle lineage[38,51], is a potential partner for myogenin at loci unique to control (Fig. 2). However, myogenin loci unique to rexinoids harbor a double E-box, as well as an overall increase in the number of E-box motifs per locus (Fig. 2). Casey and colleagues[52] have suggested that an increase in E-box density may be important for improving DNA accessibility in the context of a closed chromatin conformation, where based on the length of spacing, a series of motifs may serve as a platform on the surface of a nucleosome, to promote bHLH factor binding. Thus, the presence of secondary E-box in close proximity to the primary E-box, suggests that a bHLH dimer as well as E-box density may both play roles in increasing the binding intensity of myogenin in rexinoid-enhanced myoblast differentiation.

Interestingly, the presence of multiple E-box motifs on adjacent helical turns can create a platform permissible for a tetrameric bHLH complex. Chang et al.[39] recently illustrated the formation of an intramolecular tetramer within the TWIST family of bHLH transcription factors with a spatial configuration of a double E-box separated by five nucleotides. The spacing allows a full turn of the DNA double helix such that the E-boxes are facing the same spatial direction, with minimal steric hindrance between the E-boxes, providing efficient co-factor interactions to promote enhancer function[39]. Furthermore, we have observed a difference in the length of peaks associated to single or double E-box where the latter contained broader span of read signals (Fig. 2), which may translate to functional relevance as studies have shown that peak shape analysis can reveal information regarding the binding of neighboring proteins and in certain instances, is correlated with gene expression[53]. Taken together, rexinoid-promoted loci recognition and selection may contribute to the overall increase in myogenin binding intensity and capacity.

Importantly, the majority of rexinoid-responsive genes are associated with a myogenin peak, accompanied by an increase in

H4K8 and H3K9 acetylation in contrast to genes upregulated in differentiation only (Fig. 3). An overexpression of myogenin in C2C12 myoblasts has been shown to increase hyperacetylation of H4 associated directly to late muscle genes, such that myogenin induces chromatin remodeling of MyoD-initiated late muscle genes, and shifts the transcriptional activation of these genes to an earlier time period[13]. Since RXR signaling increases the level of myogenin protein[10], it may promote loci-specific histone acetylation of late muscle genes, leading to an earlier expression of rexinoid-responsive genes and thus enhance the differentiation and fusion of myoblasts.

We have also examined the genomic dynamics of p300, a HAT crucial for myogenic differentiation, due to the coupling of histone acetylation with rexinoid-responsive gene expression. While p300 and its functional homologue CBP share many functional parallels, they have distinct cellular functions[54–56]. Severe myogenic abnormalities are found in p300 knockout mice, but not in CBP knockouts[55,57]. In addition, ES cells from p300 knockout mice are unable to express MyoD or Myf5, whereas CBP[−/−] ES cells can develop into multinucleated myotubes, demonstrating that compared with CBP, p300 is required for muscle development[41].

Interestingly, we have found that p300 is mainly associated with active promoters in response to RXR signaling (Fig. 4), contrasting to its large association to enhancers in the absence of rexinoids[12]. Also, p300 may be recruited to the promoters by Atf1, Sp1 and bZIP binding proteins, where Atf1 motif displayed the largest enrichment at p300 associated promoters unique to rexinoid (Fig. 4). Combinations of Jun, Fos and Atf heterodimers are part of a large family known as the activating protein-1 (AP-1) transcription complex, in that the composition of AP-1 dimers, their relative abundance, and cell type determine cellular fate outcome[58]. Motif analysis of p300-associated promoters suggests recruitment of p300 by AP-1 family members during rexinoid-enhanced differentiation, with a predisposition by Atf1 binding protein in particular (Fig. 4). On the other hand, Sp1 motif displays the greatest enrichment at promoters unique to control (Fig. 4). Many studies have shown the interaction of Sp1 with p300 within the same regulatory complex on promoters of genes which they regulate, such as p21[59], IL-12[60], and IQGAP[61]. The presence of p300 allows for assembly of the transcription complex in which it acts as a bridging factor to connect transcription factors to the basal transcription machinery, but also leads to targeted histone acetylation to open up the chromatin environment and to ¨bookmark¨ the promoters for rapid transcription

re-initiation. As such, the promoters of rexinoid-responsive genes exhibit a significant increase in H3K9 acetylation, which is not the case for promoters of genes upregulated in differentiation only (Fig. 5). More importantly, knockdown of p300 diminishes the expression of myomixer, a rexinoid-responsive myogenic target (Fig. 5). Therefore, RXR signaling is exemplified by the association of p300 to promoters and an enrichment in H3K9 acetylation in particular (Fig. 5). Taken together, p300 function as a scaffold for protein complexes and a HAT for residue-specific histone acetylation at promoters may together facilitate rexinoid-responsive gene expression.

Apart from promoters, enhancers also play a key role in determining cell-type specific gene regulation, as the chromatin marks at enhancers are cell-type-specific and a functional signature of cellular fate[62–64]. We have identified binding motifs for Ap-1, Runx and Tead4 within p300 associated enhancers (Fig. 4), all of which are known myogenic enhancer regulators[51,65,66]. However, p300 may largely be recruited to the enhancer regions by E-box binding proteins (Fig. 4), following RXR signaling where a considerable overlap between p300 and myogenin loci occurs (Fig. 6). Enhancers co-occupied by p300 and myogenin are also marked by an enrichment of H4K8 and H3K9 acetylation (Fig. 6), such that the recruitment of p300 by E-box binding proteins and residue-specific histone acetylation may be a signature of rexinoid-responsive gene expression (Fig. 7).

In conclusion, we have provided molecular insights into the role of myogenin-mediated RXR signaling in skeletal myoblast differentiation (Fig. 7). A thorough understanding of the epigenetic mechanisms that regulate myogenic fate transitions will allow the development of novel therapeutics to treat muscle-related diseases, as the ability to control endogenous epigenetic modifiers will enable specific changes in transcriptional activity of genes important for myogenesis.

## Methods

**Cell culture**. C2C12 mouse myoblasts (ATCC) were maintained in growth medium (GM), consisting of Dulbecco's modified Eagle medium (DMEM) supplemented with 10% fetal bovine serum, 100 units/ml penicillin and 100 μg/ml streptomycin, at 37 °C with 5% CO$_2$. To induce differentiation, cells at 70–80% confluence were switched to differentiation medium (DM), DMEM supplemented with 2% horse serum, and cultured for 24 h. Bexarotene (LC Laboratories) was added to DM at a final concentration of 50 nm for treatment.

**Western blot analysis**. Cells were lysed in whole cell extract buffer (10% glycerol, 50 mM Tris-HCl pH 7.6, 400 mM NaCl, 5 mM EDTA, 1 mM DTT, 1 mM PMSF, 1% NP-40), at 4 °C for 30 min with rotation. Cell extracts were prepared as previously described[67]. Protein concentrations were determined by the Bradford Method (Bio-Rad). The proteins were then separated by SDS-PAGE and transferred to an Immun-Blot PVDF membrane. Bio-Rad ChemiDoc MP System was used to capture chemiluminescent images, and the protein bands were quantified using Scion Image (Scion Corporation). The primary antibodies used for western analysis were from hybridoma E7 for β-tubulin and hybridoma F5D for myogenin.

**Reverse transcription qPCR analysis**. Total RNA from C2C12 cells was isolated by using the GeneJET RNA Purification Kit (ThermoFisher Scientific) and quantified by Nanodrop (ND-1000). An equal amount of RNA was reverse transcribed into cDNA by using a High Capacity cDNA Reverse Transcription Kit (Applied Biosystems) in accordance with the manufacturer's instructions. SYBR® Green PCR Master Mix and HotStarTaq DNA polymerase (Qiagen) were used for qPCR on a CFX96 Touch Real-Time PCR Detection System (BioRad). The data was analyzed by using the threshold cycle comparative method, normalized to the expression of TATA-Binding Protein (TBP), using the formula $2^{-\Delta\Delta CT}$. Experiments were repeated at least three times. The following primers were used: myomixer primer forward, GTTAGAACTGGTGAGCAGGAG and reverse, CCATC GGGGAGCAATGGAA; Mybph primer forward, ACTTAGCCACCACCACCAAG and reverse, GGAGTGGAGGTATGGTCAGC; TBP primer forward, TCATGGA CCAGAACAACAGC, and reverse, GCTGTGGAGTAAGTCCTGTGC.

**ChIP-seq and data processing**. Following 24 h of differentiation, cells were fixed, crosslinked, and sonicated with a Bioruptor as previously described[68]. Chromatin immunoprecipitation was performed with antibodies obtained from Santa Cruz

against myogenin (sc-12732) and p300 (sc-584). The immunoprecipitants were purified using a MiniElute PCR Purification kit (Qiagen). ChIP-seq libraries were prepared and sequenced on an Illumina HiSeq 2000 (p300 and H3K27me3) or Illumina HiSeq 4000 (myogenin) as single-end 50 bp reads. For p300 and H3K27me3, reads were mapped to the mm9 genome using Bowtie[69] v0.12.7 allowing for three mismatches and reporting the single best alignment per 50 bp read. For myogenin, sequence reads were aligned to the mouse mm9 reference genome build using BWA-Mem[70] v0.7.10 with default parameters. Duplicate reads were removed utilizing the Picard package v2.6.0. Genome-wide read coverage and enrichment was assessed using deepTOOLS[71] fingerprint plots where the corresponding x- and y-axes for a point of intersection, denote the percentage of genome coverage and the percentage of all uniquely aligned reads, respectively. Peak calling was performed with HOMER[72] v4.10 with -style factor and FDR of $1.0 \times 10^{-4}$. The approximate IP efficiency calculated by HOMER describes the fraction of tags found in peaks compared with the genomic background and is utilized as one measure of ChIP efficacy.

De novo motif discovery was performed with MEME using the "meme-pal" option with input sequences of 500 bps surrounding the peak center. The mergePeaks tool from HOMER was utilized for co-localization of peaks (-d given) and the findMotifsGenome tool for de novo motif discovery on the 200 bp surrounding the peak centers. The annotatePeaks.pl script was employed for quantification of motif enrichment and for peak annotation, where HOMER compares the distance from a peak to the RefSeq catalog of TSS of nearby genes, and reports the closest TSS as the associated gene.

The enrichment of histone acetylation at specified loci or TSSs was calculated with ngs.plot[73] which calculates the coverage vectors for each query region based on specified alignment files. Coverage data is normalized for equal length and vectors are normalized against the corresponding library size, to allow comparisons of datasets with differences in sequencing depth. Default values for all analyses were used unless specified. IGV v2.3.97 was used for data browsing and representative snapshots.

**Quantitative ChIP analysis**. Following ChIP, qPCR was performed in triplicate reactions using the SYBR green method with locus specific primers. A standard curve was created for each set of primers with input DNA, followed by quantification of the abundance of immunoprecipitated target DNA in relation to input chromatin DNA. Each qChIP was repeated at least three times. Antibodies against p300 and myogenin were the same as for Western. Primer pairs used for qPCR amplification were as follows: myomixer locus forward, GAACAGCTGTGTTCTG GCAC and reverse, TGGGGAATTCCTGCACATGG; Mybph locus forward, AGTGCCTGCTCAGCTAATCC and reverse, CTGAGCCTCCTAAGCAGCAA.

**Statistics and reproducibility**. All statistical analyses were performed using The R Project for Statistical Computing or Microsoft Excel. Normally distributed data were analyzed by a two-tailed Student's t-test. Non-normally distributed data were analyzed by non-parametric two-sided Wilcoxon rank sum tests. A $p < 0.05$ was considered statistically significant. Statistical details and number of replicates are indicated in the figure legends.

**Reporting summary**. Further information on research design is available in the Nature Research Reporting Summary linked to this article.

## Data availability

All ChIP-seq datasets have been deposited in the NCBI Gene Expression Omnibus (GEO) under accession number GSE139942. Source data are available as Supplementary Data 1. All other relevant data supporting the key findings of this study are available within the article and its Supplementary Information or from the corresponding author upon reasonable request.

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

## Acknowledgements
This research was supported by an Operating Grant from the Natural Sciences and Engineering Research Council of Canada (NSERC #250174 to Q.L.). S.K. is a recipient of Canada Graduate Scholarship from the NSERC, and M.H. a scholarship from Umm Al-Qura University in Saudi Arabia.

## Author contributions
S.K. and Q.L., data analysis and manuscript preparation; M.H., data collection; J.C. and Q.L. conception and final approval of the manuscript. All authors reviewed the manuscript.

## Competing interests
The authors declare no competing interests.
